

# Transcriptome sequencing of olfactory-related genes in olfactory transduction of large yellow croaker (*Larimichthy crocea*) in response to bile salts

Jiabao Hu[1,2,3], Yajun Wang[1,2,3], Qijun Le[1,2,3,4], Na Yu[1,2,3], Xiaohuan Cao[1,2,3], Siwen Kuang[1,2,3], Man Zhang[1,2,3], Weiwei Gu[1,2,3], Yibo Sun[1,2,3], Yang Yang[1,2,3] and Xiaojun Yan[1,2,3]

[1] Key Laboratory of Applied Marine Biotechnology, Ningbo University, Ministry of Education, Ningbo, China
[2] Key Laboratory of Marine Biotechnology of Zhejiang Province, Ningbo University, Ningbo, China
[3] College of Marine Sciences, Ningbo University, Ningbo, China
[4] Ningbo Entry-Exit Inspection and Quarantine Bureau Technical Centre, Ningbo, China

Corresponding authors
Yajun Wang, wangyajun@nbu.edu.cn
Xiaojun Yan, yanxiaojun@nbu.edu.cn

## ABSTRACT

Fish produce and release bile salts as chemical signalling substances that act as sensitive olfactory stimuli. To investigate how bile salts affect olfactory signal transduction in large yellow croaker (*Larimichthy crocea*), deep sequencing of olfactory epithelium was conducted to analyse olfactory-related genes in olfactory transduction. Sodium cholates (SAS) have typical bile salt chemical structures, hence we used four different concentrations of SAS to stimulate *L. crocea*, and the fish displayed a significant behavioural preference for 0.30% SAS. We then sequenced olfactory epithelium tissues, and identified 9938 unigenes that were significantly differentially expressed between SAS-stimulated and control groups, including 9055 up-regulated and 883 down-regulated unigenes. Subsequent Gene Ontology (GO) and Kyoto Encyclopedia of Genes and Genomes (KEGG) analyses found eight categories linked to the olfactory transduction pathway that was highly enriched with some differentially expressed genes (DEGs), including the olfactory receptor (*OR*), Adenylate cyclase type 3 (*ADCY3*) and Calmodulin (*CALM*). Genes in these categories were analysed by RT-qPCR, which revealed aspects of the pathway transformation between odor detection, and recovery and adaptation. The results provide new insight into the effects of bile salt stimulation in olfactory molecular mechanisms in fishes, and expands our knowledge of olfactory transduction, and signal generation and decline.

## INTRODUCTION

Bile salts are highly structurally variable in vertebrates, and can be classified into three types; C (27) bile alcohols, C (27) bile acids, and C (24) bile acids, with default hydroxylation at C-3 and C-7 (*Hofmann, Hagey & Krasowski, 2010*). They are biliary constituents derived from cholesterol that are synthesised in the liver and stored in the gall bladder

(*Haslewood, 1967*). Regulated by different transport proteins, these salts are released into the intestinal lumen through enterohepatic circulation (*Trauner & Boyer, 2003*). Because bile salts can help intestines to digest and absorb dietary lipids and fat-soluble vitamins (*Haslewood, 1967*; *Fuentes, Ribeiro & Arago, 2018*), they are included in the diet of fishes to improve growth and digestive enzyme activities (*Deshimaru, Kuroki & Yone, 1982*; *Alam et al., 2015*). Moreover, many studies on behaviour and physiology have reported that bile salts are important chemical signalling substances as well as effective olfactory stimuli in fishes, which have distinct sensitivity to different components (*Zhang, Brown & Hara, 2001*; *Døving, Selset & Thommesen, 1980*), but molecular studies have been limited.

During olfactory activity, odourant molecules released into the environment bind to olfactory-related receptors (*Kaupp, 2010*). To date, three types of receptor genes have been identified in fishes, namely olfactory receptors (ORs) (*Freitag et al., 1998*), vomeronasal receptors (VRs) (*Freitag et al., 1995*), and trace amine-associated receptors (TAARs) (*Eisthen, 2004*). Among them, OR genes play an essential role in many odor detecting activities (*Hu et al., 2017*; *Yabuki et al., 2016*; *Bird et al., 2018*). ORs encoding G protein-coupled receptors expressed in ciliated sensory neurons were previously identified in *Rattus norvegicus* (*Buck & Axel, 1991*). According to function, OR genes can be divided into two classes; class I ($\alpha$, $\beta$, $\gamma$, $\delta$, $\varepsilon$ and $\zeta$) and class II ($\eta$, $\theta$, $\kappa$ and $\lambda$) (*Niimura & Nei, 2005*; *Hoover, 2013*). In fishes, OR genes mainly belong to class I, which are believed to recognize water-soluble odours (*Freitag et al., 1998*; *Zhou et al., 2011*). ORs are member of a multigene family of G protein-coupled receptors and seven transmembrane domain proteins (*Buck & Axel, 1991*), and $G_{\alpha olf}$, one subunit of the G protein-coupled to OR, activates adenylyl cyclase in olfactory sensory cells (*Hansen, Anderson & Finger, 2004*; *Schild & Restrepo, 1998*; *Gonalves et al., 2016*). Olfactory signals are eventually transmitted to the brain via regulation of distinct factors in olfactory transduction (*Meredith, Caprio & Kajiura, 2012*).

Many recent studies have focused on the identification and expression of olfactory-related receptor genes in fishes (*Zhu et al., 2017*; *Fatsini et al., 2016*; *Cui et al., 2017*). In the present study, to increase our knowledge of gene expression in the whole olfactory transduction system in fishes following stimulation by bile salts, we identified the optimal concentration for stimulation in *L. crocea*, and performed deep sequencing of olfactory epithelium tissues using an Illumina HiSeq 2500 platform. Subsequent GO and KEGG pathway analyses identified significantly differentially expressed genes (DEGs) enriched in eight categories related to olfactory transduction pathway, and gene expression levels were confirmed for selected genes by RT-qPCR. The results indicate that bile salts have attractant effects on *L. crocea*. The findings provide new insight into effects of bile salt stimulation on olfactory molecular mechanisms in fishes, and expand our knowledge of olfactory transduction and olfactory signalling.

## METHODOLOGY

### Fish and bile salt stimulation treatments

The large yellow croakers (mean weight = $20 \pm 1.2$ g) used in the study were commercially reared at 25–27 °C in Xiangshan Bay, Zhejiang, China. All fish experiments were conducted

in accordance with the recommendations in the National Institutes of Health Guide for the Care and Use of Laboratory Animals. The Animal Care and Use Committee of Ningbo University approved the protocols.

Sodium cholates (SAS) with typical bile salt chemical structures were chosen for stimulation treatments (*Haslewood, 1967*). Four different concentrations of SAS diluted in distilled water (0.20%, 0.30%, 0.40% and 0.50%) were applied to SAS groups, while the control group (Control) was treated with by distilled water alone. SAS was released into cultured water slowly using an air stone tube (submerged in the center of the pond) equipped with a 20 mL syringe (100 individuals per group, three independent biological replications). The behavioural responses of each fish were classified as two types: biting the air stone (a positive feeding response), and swimming close to the air stone without biting (a positive movement response). We replaced the tested fish with another new fish for subsequent replications at all concentrations. Culture water was changed after every test, and each test was performed at 24 h intervals. Behaviours were recorded with a camera for 5 min, and the number of each type of response was recorded and analysed statistically by one-way analysis of variance (ANOVA) and Tukey's multiple comparison tests (SPSS, version 16.0).

The concentration that produced the highest number of behavioural responses was used for subsequent stimulation experiments, which were performed as described as above. After stimulation, we captured control group fish, and fish from SAS groups exhibiting significantly positive feeding responses, and immediately extracted olfactory epithelium tissues by cutting the nostrils. Olfactory epithelium tissues from 15 randomly selected fish were extracted and pooled into three 1.5 mL RNAase-free tubes (three independent biological replicates for each group) and stored in liquid nitrogen for RNA-seq and RT-qPCR experiments.

## RNA isolation, library construction and Illumina sequencing

Total RNAs were extracted using TRIzol reagent (Invitrogen, Carlsbad, CA, USA). RNA was monitored on 1% agarose gels, RNA purity was checked using a NanoPhotometer spectrophotometer (IMPLEN, Westlake Village, CA, USA), RNA concentration was measured using a Qubit RNA Assay Kit with a Qubit 2.0 Fluorimeter (Life Technologies, Carlsbad, CA, USA), and RNA integrity was assessed using an RNA Nano 6000 Assay Kit with a Bioanalyzer 2100 system (Agilent Technologies, Santa Clara, CA, USA).

Sequencing libraries were generated using an NEBNext Ultra RNA Library Prep Kit for Illumina (NEB, Ipswich, MA, USA) and barcodes were added to attribute sequences to each sample. Clustering of the barcoded samples was performed on a cBot Cluster Generation System using a TruSeq PE Cluster Kit v3-cBot-HS (Illumina). After cluster generation, library preparations were sequenced on an Illumina HiSeq 2500 platform and paired-end reads were generated.

## Assembly of sequencing data and gene annotation

Raw data were firstly processed through in-house perl scripts, and clean data were obtained by removing reads containing adapters or poly-N sequences, and reads of low quality.

Q20, Q30 and GC values were calculated, and all downstream analyses were based on high-quality clean data.

The reference genome of the large yellow croaker was downloaded from the National Center of Genome Research website (https://www.ncbi.nlm.nih.gov/genome/?term=JPYK-00000000) (*Ao et al., 2015*), and data were mapped using TopHat (version 2.0.12) and Bowtie2 (*Trapnell, Pachter & Salzberg, 2009*; *Langmead et al., 2009*). Unigenes were searched using BLASTX against the National Center for Biotechnology Information (NCBI) non-redundant protein sequence (NR) database, the NCBI non-redundant nucleotide sequence (NT) database, and Gene Ontology (GO), KEGG Orthology (KO) and SwissProt databases with an *E*-value threshold of $10^{-5}$.

## Identification of differentially expressed genes (DEGs) and functional analysis

Differential expression analysis was performed using the DEGSeq R package (1.20.0) and Reads per Kilobase Millon Mapped Reads (RPKM) values (*Mortazavi et al., 2008*). The resulting *p*-values were adjusted using the Benjamini and Hochberg's approach for controlling the false discovery rate. DEGs were selected with the criteria adjusted *p*-value <0.05 and |log2fold-change| >1.

GO enrichment analysis of DEGs was implemented by the GOseq R package, and KEGG enrichment was used to identify putative functions and pathways of DEGs (http://www.genome.jp/kegg/).

## Real-time quantitative PCR (RT-qPCR) analysis

Total RNA was reverse-transcribed into cDNA using a PrimeScript RT Reagent Kit (TaKaRa, Dalian, China). Primers were designed using Primer 5.0 software (Table 1). $\beta$-actin served as an internal normalisation control for RT-qPCR analysis, and reactions contained 2 μl cDNA, 1 μl forward and reverse primers, 10 μl SYBR Green I Master Mix (TaKaRa), and 6 μl water. RT-qPCR was performed on an Eppendorf PCR machine (Mastercycler ep Realplex, Hamburg, Germany) with one cycle at 95 °C for 2 min, followed by 40 cycles at 95 °C for 15 s, 58 °C for 15 s, and 72 °C for 20 s. The relative expression level was calculated using the $2^{-\Delta\Delta CT}$ method, and statistical analysis was performed using independent sample t-tests (SPSS, version 16.0; Armonk, NY, USA).

# RESULTS

## Selecting the optimal concentration of bile salts and assessing fish responses

For bile salt stimulation treatments, SAS was diluted four different concentrations, added slowly to water, and *L. crocea* responses were monitored (Fig. 1). For feeding responses, the fish reacted most obviously to 0.30% SAS (17.67 ± 0.58 fish responded in 5 min). Meanwhile, for movement responses, they exhibited optimal attraction responses to 0.30% SAS (64.33 ± 3.51 fish responded in 5 min) and 0.40% (48.33 ± 3.51 fish responded in 5 min). Thus, we chose 0.30% SAS for subsequent RNA-seq and RT-qPCR experiments.

**Table 1  Primers for real-time quantitative PCR (RT-qPCR).**

| Gene name | Gene ID | Primer sequence (5′ → 3′) |
|---|---|---|
| OR 2D3 | gi\|734643370\| | F: CTATGCCAGCACTCTCTTTC |
| | | R: ACAAGGTGGAGGTGAGAA |
| CALM | gi\|698455748\| | F: AGGGTGTTCATTGGTGCTCG |
| | | R: ATGTAAAGCCCACGACTCAA |
| ADCY3 | gi\|734633255\| | F: AACCCATCGTTTCCTAATCC |
| | | R: GCCGCTCTGTTTCTCCTTCT |
| GNAL | gi\|734649985\| | F: AGCATCGCTCCGCTTTC |
| | | R: ATCCCGCTGACCTCCTACA |
| CAMK2 | gi\|734594146\| | F: AATGCCACCAACGACGAG |
| | | R: TCCACCAGGTTTCCCAGA |
| CNGA | gi\|734644355\| | F: AAGTGTTTAGCCCTGGAGATTAC |
| | | R: CCGCTTTACTGCCCTTGATA |
| PKA | gi\|734635100\| | F: AACCCATCGTTTCCTAATCC |
| | | R: GCCGCTCTGTTTCTCCTTCT |
| CNGB1 a | gi\|734611524\| | F: GTGTACGACGTAGCCACGAT |
| | | R: TGAGATTCCACTGAGCGATT |
| CNGB1 b | gi\|554826775\| | F: ACTTTGTTGGTGTCTTTGCTTT |
| | | R: TCTCGGGGGATGTTGTAGG |

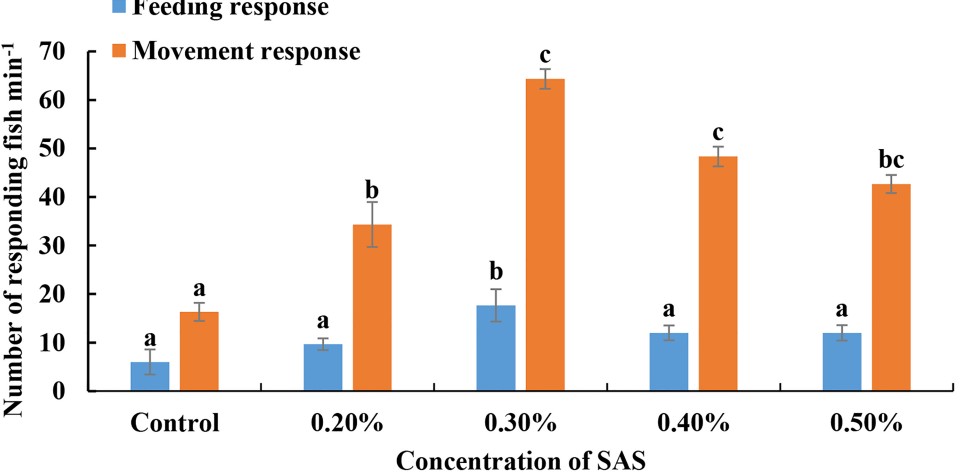

**Figure 1  Fish responses to increasing concentrations of sodium cholates (SAS).** The number of fish displaying feeding (blue bars) or movement (orange bars) responses were quantified. Fish exhibited optimal behavioural performance in response to 0.30% SAS.

## Results and analysis of transcriptome sequencing data

cDNA libraries were constructed from control and SAS groups, resulting in 39,805,502 and 39,116,990 raw reads, and more than 81% raw reads were filtered to yield clean reads. In total, 25,684,902 and 25,830,011 clean reads were mapped to the reference genome of

**Table 2** The sequence quality and mapping results between the SAS and Control groups.

| Library | Control | SAS |
|---|---|---|
| Raw Reads Number: | 39,116,990 | 39,805,502 |
| Raw Reads Length (bp): | 125 | 125 |
| Clean Reads Number: | 32,205,388 | 32,272,020 |
| Clean Reads Length (bp): | 125 | 125 |
| Clean Reads Rate (%): | 82.33 | 81.07 |
| Mapped Reads: | 25,684,902 | 25,830,011 |
| Mapping Rate(%): | 80 | 80 |
| Raw Q30 Bases Rate (%): | 90.57 | 90.23 |
| Clean Q30 Bases Rate (%): | 95.83 | 96.02 |

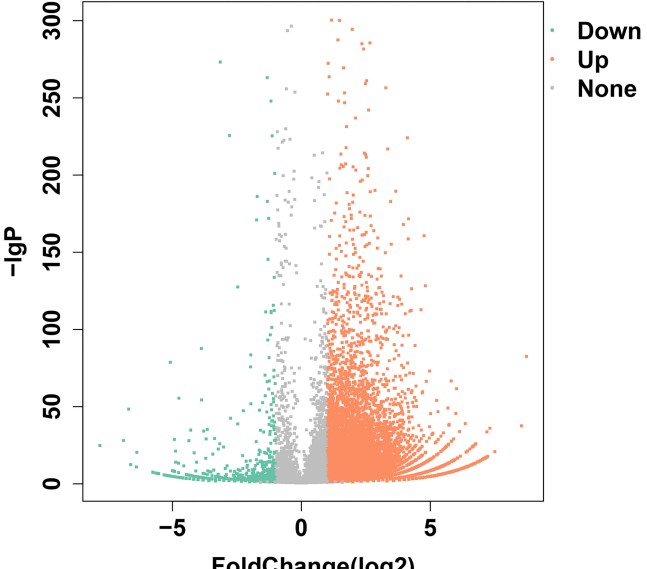

**Figure 2** **DEGs identified following bile salt stimulation.** Red spots represent up-regulated genes and green spots indicate down-regulated genes. Grey spots represent genes showing no obvious change between control and SAS groups.

*L. crocea* for control and SAS groups, respectively, and the Q30 value was >95% for libraries (Table 2).

## Identification and functional annotation of DEGs

Transcriptome data from olfactory epithelium tissues of control and SAS groups were compared, and 19,197 unigenes were annotated, of which 9938 DEGs met the criteria ($|\log2\text{Foldchange}| > 1$ and $p < 0.05$). Of these, 9055 were up-regulated and 883 were down-regulated (Fig. 2). Three types of olfactory-related receptor genes were found to be differentially expressed in our data (all up-regulated), comprising 59 *ORs*, two *VRs* and 17 *TAARs*.

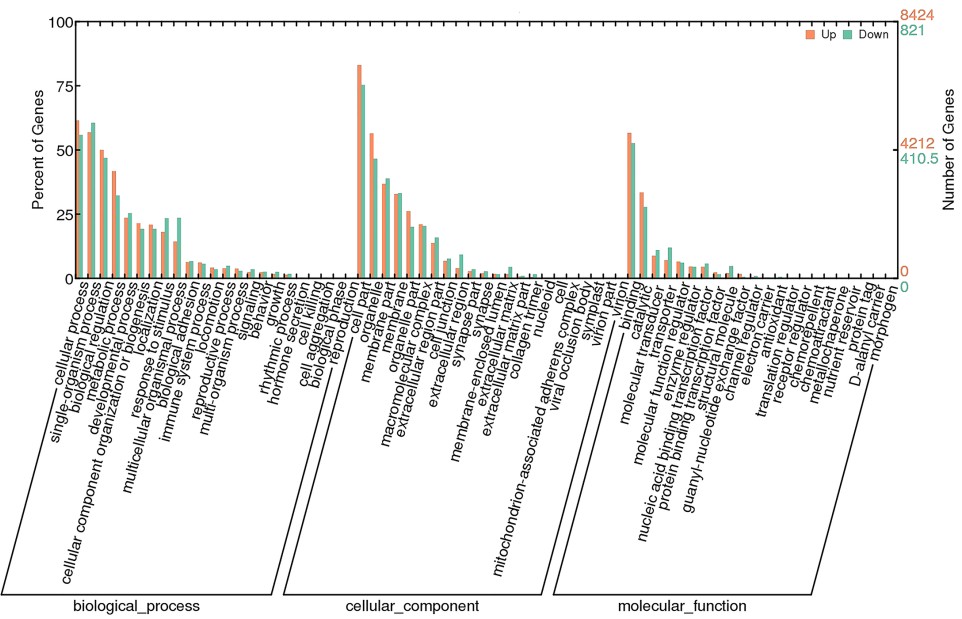

**Figure 3** **GO analysis of DEGs identified by comparing control and SAS groups.** Orange represents up-regulated genes and green indicates down-regulated genes. The height of bars is proportional to the number of DEGs.

To investigate the functions of DEGs, 9245 unigenes (8424 up-regulated and 821 down-regulated) were assessed in terms of the three main GO classifications, namely biological processes (BP), cellular component (CC), and molecular function (MF; Fig. 3). According to the criteria ($p$-value < 0.001), single-organism process (GO:0044699), intrinsic to membrane (GO:0031224) and substrate-specific channel activity (GO:0022838) were highly represented, and potentially played an important role in olfactory responses to bile salts.

To identify KEGG pathways between control and SAS groups, 3,140 DEGs were mapped to 321 pathways, and 20 pathways were highly enriched according to $q$-value <0.05 (Table 3). Among these pathways, olfactory transduction (map04740) was mainly involved in olfactory responses to bile salts. In this pathway, 73 differentially expressed olfactory-related genes were enriched among eight categories including calmodulin (CALM, k02183), adenylate cyclase 3 (ADCY3, k08043), guanine nucleotide-binding protein G (olf) subunit alpha (GNAL, k04633), calcium/calmodulin-dependent protein kinase (CaM kinase) II (CAMK2, k04515), olfactory receptor (OLFR, k04257), cyclic nucleotide gated channel beta 1 (CNGB1, k04952), cyclic nucleotide gated channel alpha 3 (CNGA3, k04950) and protein kinase A (PKA, k04345; Fig. 4). Significantly differentially expressed olfactory-related genes in these categories were subsequently analysed RT-qPCR (Table 4).

In the olfactory transduction pathway (Fig. 4), olfactory stimulation could be divided into odor detection, and recovery and adaptation. During odor detection, odour-activated OLFR stimulates G protein release protein GNAL, and ADCY3, which is positively regulated by GNAL, high concentration cAMP activates CNGB1, leading to the entrance of $Na^+$

**Table 3   KEGG pathway analysis of the 20 highly enriched categories.**

| Pathway ID | q-value | Pathway |
|---|---|---|
| map03010 | 3.77E–15 | Ribosome |
| map05012 | 1.59E–08 | Parkinson's disease |
| map03008 | 1.01E–07 | Ribosome biogenesis in eukaryotes |
| map03040 | 3.40E–07 | Spliceosome |
| map03050 | 1.29E–06 | Proteasome |
| map03013 | 1.72E–06 | RNA transport |
| map00190 | 1.72E–06 | Oxidative phosphorylation |
| map03030 | 1.94E–06 | DNA replication |
| map03430 | 4.59E–05 | Mismatch repair |
| map00970 | 0.000145129 | Aminoacyl-tRNA biosynthesis |
| map05016 | 0.001811227 | Huntington's disease |
| map04740 | 0.004459478 | Olfactory transduction |
| map03440 | 0.005177196 | Homologous recombination |
| map03420 | 0.005828765 | Nucleotide excision repair |
| map04721 | 0.008139829 | Synaptic vesicle cycle |
| map04920 | 0.029436551 | Adipocytokine signaling pathway |
| map04142 | 0.037198669 | Lysosome |
| map05160 | 0.037198669 | Hepatitis C |
| map04111 | 0.041314855 | Cell cycle - yeast |

and $Ca^{2+}$ into olfactory sensory cells. This process is an example of signal production and amplification. During recovery and adaptation, an increase in cAMP activates PKA, which phosphorylates OLFR; meanwhile, CALM represses CNGB1 and activates CAMK2 to suppress ADCY3 by phosphorylation. This process represents an example of signal suppression.

### RT-qPCR analysis of eight categories related to olfactory transduction

The expression levels of nine DEGs related to the olfactory transduction pathway were validated by RT-qPCR. These genes were all significantly expressed in the olfactory epithelium (*$p < 0.05$ and **$p < 0.01$), especially *CAMK2*, *ADCY3*, *OR 2D3* and *CNGB1*, confirming the reliability of the transcriptome sequencing data (Fig. 5). Furthermore, *CNGB1 a* and *CNGB1 b* both belonging to *CNGB1*, and *CNGB1 a* (up-regulated) displayed more significant differential expression than *CNGB1 b* (down-regulated).

## DISCUSSION

### Bile salts act as effective olfactory stimuli in fishes

Fish can display different olfactory responses to different levels of odours. Using electroolfactograms (EOGs), many fish species have been shown to possess specific thresholds to different levels of cholic acid (CA) (*Meredith, Caprio & Kajiura, 2012*; *Døving, Selset & Thommesen, 1980*). In the present study, we found that *L. crocea* acted differently to different concentrations of bile salts based on behavioural analysis, and responded

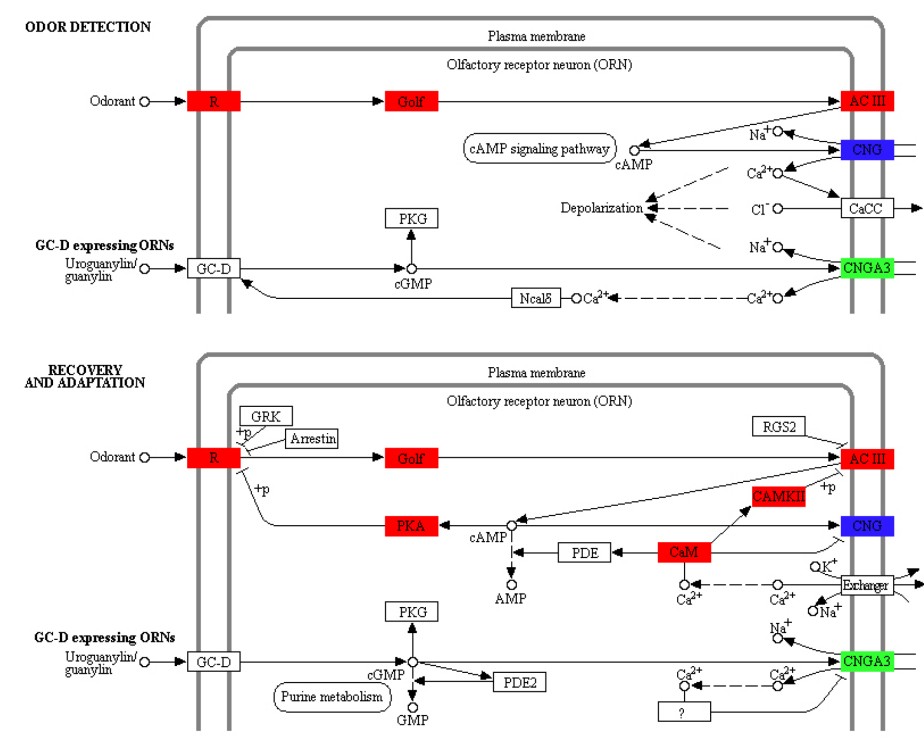

**Figure 4 Significantly differentially expressed genes in eight categories related to the olfactory transduction pathway.** Red indicates significantly up-regulated genes, green indicates significantly down-regulated genes, and blue indicates genes that were both up- and down-regulated. (Image credit: https://www.kegg.jp/kegg-bin/show_pathway?map04740).

**Table 4 Nine highly differentially expressed genes in 8 categories from olfactory transduction.**

| Gene name | log$_2$FoldChange | $P$-value | Gene ID | Description |
|---|---|---|---|---|
| *OR 2D3* | 5.388277756 | 1.13E–11 | gi\|734643370\| | Olfactory receptor 2D3 |
| *CALM* | 2.148730614 | 1.35E–25 | gi\|698455748\| | Calmodulin |
| *ADCY3* | 4.803315255 | 4.82E–05 | gi\|734633255\| | Adenylate cyclase type 3 |
| *GNAL* | 1.303754743 | 9.90E–96 | gi\|734649985\| | Guanine nucleotide-binding protein G(olf) subunit alpha |
| *CAMK2* | 3.964251474 | 6.05E–76 | gi\|734594146\| | Calcium/calmodulin-dependent protein kinase type II subunit gamma |
| *CNGA* | −1.827161905 | 3.96E–06 | gi\|734644355\| | Cyclic nucleotide-gated channel cone photoreceptor subunit alpha |
| *PKA* | 2.004522347 | 1.89E–19 | gi\|734635100\| | cAMP-dependent protein kinase catalytic subunit PRKX |
| *CNGB1 a* | 4.165885335 | 1.29E–47 | gi\|734611524\| | Cyclic nucleotide-gated cation channel beta-1 |
| *CNGB1 b* | −4.004039667 | 0.001710637 | gi\|554826775\| | Cyclic nucleotide-gated cation channel beta-1 |

optimally to 0.30% SAS rather than to higher levels. We believe that fishes have limited olfactory-related receptors, which leads to limited olfactory ability, explaining why they do not exhibit significantly more intense behaviour with increased levels of odours. This

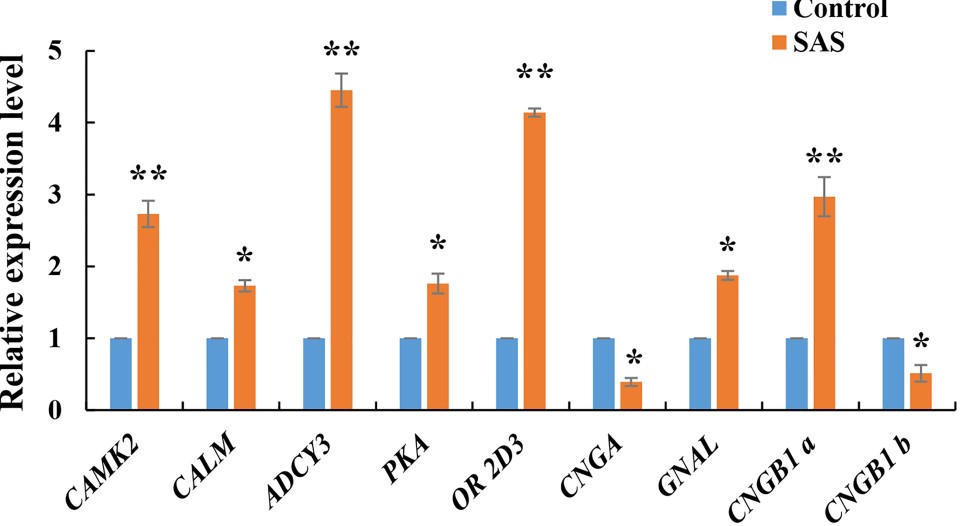

**Figure 5   Relative expression levels of nine DEGs related to olfactory transduction.** The results were calculated according to the $2^{-D\Delta CT}$ method using β-actin as an internal reference gene. * $p < 0.05$ and ** $p < 0.01$.

suggests that fishes may possess a maximum detection peak for concentrations of certain stimuli, and a similar phenomenon has been observed in other studies (*Zhao, 2007*; *Hu et al., 2017*).

The olfactory sensitivity of fishes can also be measured using EOGs. However, unlike EOG analysis, our behavioural experiments revealed fish response to stimuli directly (approach and avoidance). By imitating conditions in which fishes detect bile salts in natural environments, we found that *L. crocea* performed feeding movements upon exposure to SAS, which suggests that the fish had a particular preference toward SAS. Indeed, many studies have demonstrated that some bile salts could be good phagostimulants for fish feed (*Hu et al., 2017*; *Rolen & Caprio, 2008*; *Yamashita, Yamada & Hara, 2010*), suggesting that they may act on both olfactory and taste pathways in fish, and might be good attractants.

## Analysis of olfactory-related receptor genes in the odorant transduction cascade

In *L. crocea*, OR gene family is the largest one of three olfactory-related receptor genes families (*Ao et al., 2015*; *Zhou et al., 2011*). In our current study, 59 OR genes were found to be all up-regulated after the fish were stimulated, many more than two other two types of receptors, consistent with previous studies (*Saraiva & Korsching, 2007*; *Hashiguchi & Nishida, 2006*; *Hu et al., 2017*). Thus, ORs appear to be the major receptors responding to bile salts in *L. crocea*. Fish produce and release bile salts as sex pheromones to communicate with other individuals (*Zhang, Brown & Hara, 2001*). However, regarding pheromone receptors in the epithelium (*Muramoto et al., 2011*), only two VR genes were differentially expressed (up-regulated) in the present study. We speculate that the fish used in our study might be juveniles, hence VRs were not sensitive to sex pheromones at this stage of the life

cycle. Moreover, 17 TAAR genes were found to be all up-regulated following stimulation by SAS in our study. Interestingly, TAARs could only be activated by amines at trace level in a previous study (*Borowsky et al., 2001*), and SAS is not an amine, suggesting that the fish might release some amines substances to communicate with each other in response to SAS. Our study reveals that ORs might be the main bile salt receptors in the olfactory epithelium during different developmental stages in fish species.

## Signal transduction and regulation components

During signal transduction, ORs bind to their corresponding G proteins, among which G $\alpha$ is one of most important subunits (*Jones & Reed, 1989*). In the present study, $G_{\alpha olf}$ was released in the olfactory transduction pathway after ORs were activated by SAS, and two $G_{\alpha olf}$ genes were up-regulated alongside high expression of *ORs*, which suggests that olfactory receptors bind to G protein possessing the $G_{\alpha olf}$ subunit. Some studies on olfactory sensory neurons also have confirmed similar binding relationships of them (*Jones & Reed, 1989*; *Ronnett & Moon, 2002*). However, only three $G_{\alpha olf}$ genes were identified in *L. crocea*, indicating that they may be a small gene family in this fish species.

In the present study, the $G_{\alpha olf}$ subunits activated ADCY3, which led to a rise in cAMP levels during olfactory transduction, which is of clear relevance to signal transduction (*Jones & Reed, 1989*; *Dhallan et al., 1990*; *Menco et al., 1992*). We also found that ADCY3 was enriched among up-regulated genes in the pathway, which suggests that this factor could act positively on signal transduction, and play a key role in regulating transformation of the pathway via the cAMP levels. Moreover, ADCY3 was the first factor in secondary signal transduction (Fig. 4), and some other studies have reported that signal transduction can be disrupted if ADCY3 genes are mutated (*Brunet, Gold & Ngai, 1996*; *Hacker, 2000*). Thus, our results indicate that ADCY3 is one of most important factors mediating signal transduction between primary and secondary signal transduction.

In odor detection of olfactory transduction, high cAMP levels produced by ADCY3 activated CNGB1, leading to the entrance of $Na^+$ and $Ca^{2+}$ into olfactory sensory cells. Other studies have reported similar results (*Michalakis et al., 2006*; *Kaupp & Seifert, 2002*). Thus, we speculate that an increase in these two ions by CNGB1 might appeared to suppress the expression of *CNGA3* (*Wissinger et al., 2001*), another same functional channel protein-encoding genes, due to competition effects. In recovery and adaptation of the pathway, activated CALM resulting from increased $Ca^{2+}$ regulated by CAMK2 suppresses CNGB1, leading to a drop in cAMP level indirectly, consistent with observations in previous studies (*Cheung, 1980*; *Lynch & Barry, 1989*; *Menini, Picco & Firestein, 1995*; *Kapiloff et al., 1991*). Thus, a series of interactions may cause $Ca^{2+}$ levels to decline, reducing the intracellular and extracellular charge difference. In addition, *CNGB1a* was expressed at higher levels than *CNGB1b* based on RT-qPCR results, which indicates that signal transduction in fish might be transforming odor detection into recovery and adaptation. These findings reveal that *CNGB1a* and *CNGB1b* may be involved in signal production and decline in the pathway, respectively.

PKA can help ORs to bind to G proteins (*Daaka, Luttrell & Lefkowitz, 1997*; *Zamah et al., 2002*) in a mechanism mediated by cAMP (*Chang, Yu-Ming & Zhang, 2006*), and our

results showed that *PKA* (up-regulated) was enriched during the recovery and adaptation aspect of olfactory transduction. Thus, PKA might suppress the initial signal level by hindering the separation between receptor and G protein subunit by phosphorylation. Combined with the results of a previous study (*Taiwo et al., 1989*), our findings indicate that the inhibitory action of PKA may be activated through a change in cAMP concentration due to binding between ORs and G proteins, and indirectly by suppression of ADCY3. These factors might alter the electric charge in olfactory sensory cells by meditating the ion concentration, which might lead to changes in electric signalling between olfactory receptor cells and olfactory sensory neurons, consistent with some previous reports (*Menini, Picco & Firestein, 1995*; *Lynch & Barry, 1989*). Our results therefore indicate that odor detection in fish may occur quite rapidly, or a long time after, stimulation by bile salts, and recovery and adaptation may occur once fish become familiar to this stimulation.

## CONCLUSION

In the present study, *L. crocea* displayed a significant behavioural preference for 0.3% SAS, which could be a good attractant in fishes. We performed transcriptome sequencing of olfactory epithelium tissues to identify olfactory-related genes involved in the olfactory transduction pathway, and eight categories were found to be highly enriched with DEGs in related DEGs, especially *CAMK2*, *ADCY3*, *OR 2D3* and *CNGB1*. The pathway could be divided into two processes: odor detection, and recovery and adaptation, and involves DEGs such as *CAMK2*, *CALM*, *CNGB1* and *PKA* that may regulate conversion between the two processes. Our results provide new insight into the effects of bile salt stimulation on olfactory molecular mechanisms in fishes, and expand our knowledge of olfactory transduction and signal production and decline.

## ACKNOWLEDGEMENTS

We thank Shunshun Tao of the Xiangshan Harbor Aquaculture and Larva Limited Company and Dr. Bao of Wanli College of Zhejing.

### Funding

This work was supported by the Key projects of the Ningbo people's livelihood in Agriculture (2013C11010), the Natural Science Foundation of China (31772869 and 31872586), the Natural Science Foundation of Zhejiang (LY18C190008 and LY18C1900013), the Agriculture Key Special Project of Ningbo (2015C110003), Research Foundation of State General Administration of The People's Republic of China for Quality Supervision and Inspection and Quarantine (2016IK173 and 2017IK295), the Ningbo Livelihood Key Project (2013C11010), the Zhejiang Major Science Project (2019C02059), the Optimization and Utilization of Acute Toxicity Test Methods for Luminescent Bacteria (2019ZS02) and the K.C. Wong Magna Fund in Ningbo University, Li Dak Sum Yip Yio Chin Kenneth Li Marine Biopharmaceutical Development Fund, National 111 Project of

China. The funders had no role in study design, data collection and analysis, decision to publish, or preparation of the manuscript.

**Grant Disclosures**

The following grant information was disclosed by the authors:

Ningbo people's livelihood in Agriculture: 2013C11010.

Natural Science Foundation of China: 31772869, 31872586.

Natural Science Foundation of Zhejiang: LY18C190008, LY18C1900013.

Agriculture Key Special Project of Ningbo: 2015C110003.

Research Foundation of State General Administration of The People's Republic of China for Quality Supervision and Inspection and Quarantine: 2016IK173, 2017IK295.

Zhejiang Major Science Project: 2019C02059.

Optimization and Utilization of Acute Toxicity Test Methods for Luminescent Bacteria: 2019ZS02.

K.C. Wong Magna Fund in Ningbo University.

Li Dak Sum Yip Yio Chin Kenneth Li Marine Biopharmaceutical Development Fund, National 111 Project of China.

**Competing Interests**

The authors declare there are no competing interests. Qijun Le is employed by Ningbo Entry-Exit Inspection and Quarantine Bureau Technical Centre.

**Author Contributions**

- Jiabao Hu conceived and designed the experiments, performed the experiments, analyzed the data, contributed reagents/materials/analysis tools, prepared figures and/or tables, authored or reviewed drafts of the paper, approved the final draft.
- Yajun Wang conceived and designed the experiments, contributed reagents/materials/-analysis tools, authored or reviewed drafts of the paper, approved the final draft.
- Qijun Le performed the experiments, analyzed the data, contributed reagents/material-s/analysis tools, prepared figures and/or tables.
- Na Yu, Xiaohuan Cao, Siwen Kuang, Man Zhang, Weiwei Gu, Yibo Sun and Yang Yang performed the experiments, contributed reagents/materials/analysis tools, prepared figures and/or tables.
- Xiaojun Yan conceived and designed the experiments, authored or reviewed drafts of the paper.

**Animal Ethics**

The following information was supplied relating to ethical approvals (i.e., approving body and any reference numbers):

All fish experiments were conducted in accordance with the recommendations in the National Institutes of Health Guide for the Care and Use of Laboratory Animals. The Animal Care and Use Committee of Ningbo University approved the protocols.

**Data Availability**

Raw sequencing data are available through the NCBI Sequence Read Archive (accession: PRJNA378130).

## Supplemental Information

Supplemental information for this article can be found online at http://dx.doi.org/10.7717/peerj.6627#supplemental-information.

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
