# Peer review of "Transcriptome sequencing of olfactory-related genes in olfactory transduction of large yellow croaker (Larimichthy crocea) in response to bile salts"

_PeerJ, doi:10.7717/peerj.6627_

## Round 0.1 · original submission · Major Revisions

I have heard from two reviewers, both of whom agree your manuscript has merit to eventually be published, while also noting that more work is needed on general discussion. I would particularly pay attention to the first reviewer's comments on redoing the discussion, and agree largely with their constructive comments.

In my own review, I also wish to emphasize two points. First of all, the English has numerous odd mistakes (using "&" instead of "and"; species name capitalization mistakes, for example) that need correction. Please edit the English thoroughly and provide a certificate of editing by a company, or the name of the person and email address for whomever edits the paper. Note PeerJ does not perform in-house editing as part of their production process, and so your paper MUST meet international scientific English status. I will not send the paper out for review until this point is cleared.

As well, you must clearly delineate how the current data are different from data used in this paper:

https://onlinelibrary.wiley.com/doi/abs/10.1111/are.13559

You must also clearly explain if the fish or dataset used are the same or different (generated at the same or different times), and clearly explain all your methodologies concretely. As in your previous paper above you suggest "Furthermore, improved understanding via olfactory epithelium transcriptomic analysis would inform the development of an all–plant protein diet for L. crocea." and it seems that is what you have done, I cannot understand why you fail to reference your own work.

Reviewer 1 ·

Basic reporting

Clear and unambiguous, professional English used throughout.
Must be improved (see report below)

Literature references, sufficient field background/context provided.
'no comment'

Professional article structure, figs, tables. Raw data shared.
'no comment'

Self-contained with relevant results to hypotheses.
Must be improved (see report below)

Experimental design

no comment

Validity of the findings

no comment

Additional comments

The present paper by Jiabao et al. investigated the expression of olfactory-related genes stimulated by bile salts in olfactory transduction in the fish Larimichthys crocea. The experimental level of study as well formal aspects are very good. The object of study has attracted larger research interest and present submitted study deepens the knowledge of this interesting model. As such, MS could be good contribution; however, the text itself suffers from many problems and unfortunately in present version must be rejected. This MS is a classical example of nice data lost in poorly, superficially formulated text. Both the English writing style (see comments below) and the general organization from Abstract/Introduction and particularly the discussion, which in the present form contains a lot of introduction material and looks like an expansion of the results section, more than in fact, discuss hypothesis and present a solid home message for the readers. I listed some (not necessary all) examples of incongruences in the text and suggest the authors to complete reformulate Abstract/Intro/discussion whole structure before any new attempt of submission.

Fish x fishes
Please keep in mind that Fish can refer to multiple fish individuals, especially when they are all the same species of fish. Fishes, however, usually refers to multiple species, especially in scientific contexts. Therefore, most of the places in the text the correct plural form should be Fishes.

Keywords should not repeat words from the Title. Please select a different set of.

Lines 48-57: The word “bile salts” appeared 8 times just in these sentences. Please find a different way of referring to it in order to avoid constant repetition of words (E.g: “Bile salts are biliary constituents derived from cholesterol that are synthesized in the liver and stored in the gall bladder (Haslewood, 1967). Regulated by different transport proteins, they are released into the intestinal”
Lines 56-57: This can be deleted or moved to final part of introduction, since its content is the same present in lines 69-70.
Line 85: Please provide the number of such protocols.
Lines 199-205: This paragraph can be moved to the introduction since it does not represent any discussion of your data.
In the same way, Lines 216-224 can be moved to the introduction since it does not represent any discussion of your data.
Line 232: Besides, we found that 17 TAAR genes were all up-regulated and were …..
Lines 246- 254. Again can be moved to the introduction since it does not represent any discussion of your data.

Reviewer 2 ·

Basic reporting

The text of the manuscript by Hu and colaborators is clear and the English is appropriate. There are some small mistakes that need correction: e.g. 1 "L. Crocea" needs to be changed to "L. crocea" in many parts of the manuscript; e.g. 2 "adaption" change to "adaptation". I think that the specially in the introduction some additional changes should be introduced: its somewhat ambiguous the central aim of the work - the dataset is valuable and the analysis look correct, but I would ask to provide in the introduction a better focus. For example, comparative analysis with similar studies; olfactory systems in fish etc, and the reference list is in some cases old (not that its bad but for sure other more recent studies exist). The figures and tables look appropriate (except figure 4 which looks a bit pixellated). Some sections should be more self contained: lines 231-236; lines 274-284 (not clear at all).

Experimental design

Most of the experimental design is appropriate. Yet, the authors should define better defined; what is the gap? I failed to see how the RNA seq was conducted - how many samples were sequenced? Individual sequencing or pulls? Why 5 minutes to observe the behavior?

Validity of the findings

I find the results valide but see comments above.

Additional comments

NA

---

## Round 0.2 · Minor Revisions

I have gone over the resubmission myself. In general, you have responded well to the comments of the two reviewers, and the paper is much easier to read, with a much better Discussion.

I do note some small English errors here and there, and locations where your meaning is obscure. As well, the Acknowledgements are incomplete, and the Reference section must be completely formatted and checked; there are many inconsistencies as it is now.

Thus, while your paper is now scientifically acceptable, I am returning this to you for one round of checks and formatting as these are above the level we can ask our editing office to perform. Please respond to the comments in the attached MSWord file in detail in your new rebuttal file. I look forward to seeing a resubmission.

---

## Round 0.3 · accepted · Accept

While there are a few small issues remaining (author names in capital letters or not in the References), I believe this paper is now ready to go into production as the English is now much better and up to international standards. Thank you very much for your hard work.

# #